# Adherence to Medication in Older Adults with Type 2 Diabetes Living in Lubuskie Voivodeship in Poland: Association with Frailty Syndrome

**DOI:** 10.3390/jcm11061707

**Published:** 2022-03-19

**Authors:** Iwona Bonikowska, Katarzyna Szwamel, Izabella Uchmanowicz

**Affiliations:** 1Institute of Health Sciences, University of Zielona Góra, 65-417 Zielona Gora, Poland; 2Institute of Health Sciences, University of Opole, 45-060 Opole, Poland; katarzyna.szwamel@uni.opole.pl; 3Department of Clinical Nursing, Wroclaw Medical University, 50-367 Wroclaw, Poland; izabella.uchmanowicz@umed.wroc.pl

**Keywords:** diabetes mellitus type 2, frailty syndrome, aged, treatment adherence and compliance

## Abstract

Purpose: Diabetic patients aged 65 years or older are more likely to be frail than non-diabetic older adults. Adherence to therapeutic recommendations in the elderly suffering from diabetes and co-existent frailty syndrome may prevent complications such as micro- or macroangiopathy, as well as significantly affect prevention and reversibility of frailty. The study aimed at assessing the impact of frailty syndrome (FS) on the level of adherence to medication in elderly patients with type 2 diabetes (DM2). Patients and Methods: The research was carried out among 175 DM2 patients (87; 49.71% women and 88; 50.29% men) whose average age amounted to 70.25 ± 6.7. Standardized research instruments included Tilburg frailty indicator (TFI) to assess FS and adherence in chronic disease scale questionnaire (ACDS) to measure adherence to medications. Results: The group of 101 (57.71%) patients displayed medium, 39 (22.29%)—low, and 35 (20.00%)—high adherence. As many as 140 of them (80.00%) were diagnosed with frailty syndrome. The median of the average result of TFI was significantly higher in the low adherence group (*p* ˂ 0.001) (Mdn = 9, Q1–Q3; 7–10 pt.) than in the medium (Mdn = 6, Q1–Q3; 5–9 pt.) or high adherence (Mdn = 6.00, Q1–Q3; 4.5–8 pt.) ones. The independent predictors of the chance to be qualified to the non-adherence group included three indicators: TFI (OR 1.558, 95% CI 1.245–1.95), male gender (OR 2.954, 95% CI 1.044–8.353), and the number of all medications taken daily (each extra pill decreased the chance of being qualified to the non-adherence group by 15.3% (95% CI 0.728–0.954). Conclusion: Frailty syndrome in elderly DM2 patients influenced medical adherence in this group. The low adhesion group had higher overall TFI scores and separately higher scores in the physical and psychological domains compared to the medium and high adhesion groups.

## 1. Introduction

Populations are aging and the prevalence of diabetes mellitus is increasing tremendously [1,2]. In 2019, it is estimated that 19.3% of people aged 65–99 years (135.6 million) lived with diabetes. It is projected that the number of people older than 65 years with diabetes will have reached 195.2 million by 2030 and 276.2 million by 2045 [2]. Diabetes mellitus type 2 (T2DM) is associated with complications including cardiovascular disease, retinopathy, renal failure, and peripheral vascular disease [3].

The risk of complications such as micro or macroangiopathy increases significantly after 10 years since diabetes is diagnosed and proper application of medical recommendations is crucial to prevent them [1,4]. Diabetes mellitus also leads to a significant deterioration in the quality of life due to premature disability, blindness, terminal chronic kidney disease, nontraumatic amputations, as well as frequent causes of hospitalization [5]. Recently, several data suggested that diabetes is accompanied not only by complications and disability but also frailty syndrome [4,6].

Frailty syndrome is a multidimensional clinical state that is common in older age [7]. The syndrome appears as a result of decreasing body physiological reserves and decreased the reply of the body to stressors’ reaction connected with a deteriorating capacity of physiological systems and their deregulation. It leads to walking, balance, mobility as well as cognitive ability impairments. According to this concept developed by Fried et al., frailty is “a clinical syndrome in which three or more of the following criteria were present: unintentional weight loss (10 lbs in the past year), self-reported exhaustion, weakness (grip strength), slow walking speed and low physical activity” [8]. The current estimate of physical frailty prevalence among community-dwelling older Europeans is around 15% for adults aged 65 years and over [9].

Previous studies have shown that diabetic patients aged 65 years or older were more likely to be frail than nondiabetics [10,11]. The pathogenic linkage between DM and frailty potentially includes premature senescence of organ systems in a hyperglycemic status, chronic inflammation, increased oxidative stress, advanced glycation end-product accumulation, and insulin resistance [12]. Angulo et al., claim that the co-occurrence of diabetes and frailty in older people is not surprising since these two age-related conditions share common underlying pathophysiological mechanisms [13]. In patients with diabetes, death and cardiovascular diseases are attributed to classical risk factors such as hypertension, dyslipidemia, and smoking in approximately 60% of the patients and the contributing factor for the remaining 40% is frailty [14].

According to the recommendations, each person aged 65 and older should be given a chance to undergo a reliable screening test. After the diagnosis of frailty or prefrailty, further clinical assessment of the health condition should be carried out. All persons with frailty should receive social support as needed to address unmet needs and encourage adherence to a comprehensive care plan (strong recommendation) [15].

As it might be concluded, following therapeutic recommendations is especially important in the elderly suffering from diabetes as it might not only prevent such complications as micro and macroangiopathy but it is crucial for the prevention and reversibility of frailty [7,16]. Adherence to long-term therapy is defined as” the extent to which a person’s behavior—taking medication, following a diet, and/or executing lifestyle changes, corresponds with agreed recommendations from a health care provider” [17]. It should be emphasized that about 50% of patients with chronic diseases do not have good adherence to their medication treatment plan [18]. The factors which contribute to non-adherence to therapeutic recommendations include patient-related factors (e.g., having negative beliefs and/or perceptions on medications, misconception about their medication), condition-related factors (co-morbidity, cognitive impairment, depression and anxiety, forgetfulness, alcohol consumption), socioeconomic-related factors (age > 60, lack of finances, poor health literacy) and therapy-related factors (e.g., the complexity of treatment regime and pill burden, the disappearance of symptoms, knowledge about DM and its medication) [19,20].

Medication non-adherence remains a significant barrier in achieving better health outcomes for patients with chronic diseases [21]. Following therapeutic recommendations among elderly DM patients might be even worse if frailty is co-existent. Diabetes and frailty are intricately linked; furthermore, diabetes is strongly associated with reduced mobility, activities of daily life, cognitive impairment, and dementia [22]. In recent years, researchers have repeatedly addressed the coexistence of frailty and diabetes in the elderly [23,24,25,26]. This problem was considered in the context of the impact of these diseases on the quality of life [5], cognitive and functional status [27,28,29], or on the results of clinical trials of these patients [29]. There are also studies in the literature focusing on the problem of non-adherence in various chronic diseases such as: COPD (chronic obstructive pulmonary disease [30], DM2 [31,32,33], chronic kidney disease [34], hypertension [35] or rheumatic heart disease [36], etc. There are also many studies, which show the level of adherence and effectiveness of interventions in patients with type 2 diabetes [37,38], although only some of these studies involve elderly patients. Previous studies showed that the coexistence of frailty syndrome has a negative impact on the adherence of older patients with hypertension [23] and with cardiovascular diseases [39]. However, the problem of adherence to medication in patients over 60 years of age with type 2 diabetes and coexisting frailty is rarely taken up. Analysis of our data allowed us to conclude that each score obtained on the TFI Scale increased the chance of qualifying for the non-adherence group by 55.8%. The relevant independent predictors of non-adherence qualification chance were three factors such as TFI score, male gender, and the number of all medications taken by a patient daily. We believe these results may be of great importance that should be shown and disseminated.

## 2. Materials and Methods

### 2.1. Study Design and Setting

The present study has a cross-sectional and observational design. The research was carried out between 2018 and 2019 after obtaining the consent of the Bioethics Committee at the Wroclaw Medical University (approval no. KB—207/2018) while maintaining the requirements of the Declaration of Helsinki of 1975 (amended in 2000) and Good Clinical Practice. The research was conducted among the patients of 5 primary health care centers located in Zielonogórskie District (Lubuskie Voivodeship, Poland).

### 2.2. Participants

To achieve an appropriate size for the study sample (α = 0.95), invitation letters were sent to 365 persons randomized out of 39,197 patients with diagnoses of ICD-10 (E11–E11.9) in the age range of 60–89 years old living in the Lubuskie Voivodeship in 2017. Data regarding relevant inhabitants were obtained from Lubuskie Department of the National Health Fund from 2018 [40]. Invitations to the study were sent to the managers of all 30 primary health care centers located in Zielonogórskie District (Lubuskie Voivodeship). A positive response was obtained from five of them, therefore only patients from these centers were included in the study.

Doctors identified patients for the study according to the inclusion criteria. Then, the diabetes nurse interviewed the patient, presenting the purpose and method of the study and obtaining preliminary oral informed consent. Patients received a complete set of questionnaires and a written informed consent form to participate in the study.

The research inclusion criteria included: age ≥ 60, at least a year since diabetes was diagnosed, a written consent for the participation in the study and physical ability to fill in the questionnaire. The exclusion criteria included: neoplastic disease, chronic heart failure (NYHA IV), acute respiratory disease, ischemic heart disease (CCS-IV), end-stage renal disease or uremia, exacerbation of any diabetes complication making it impossible to complete the questionnaire (e.g., difficulties with reading the text of the questionnaire reported by the patient) and lack of a written consent for the participation in the study. Initially, 200 patients were accepted for the study but, eventually 175 T2DM patients were analyzed. The average age was 70.25 (SD = 6.7). The choice of the sample group is illustrated in Figure 1.

Before the examination, each patient was informed about its aim, method, and possibility of withdrawal at any stage of the research. The examinees were also assured full anonymity and the voluntary nature of the study. The aim and the procedures were explained during the selection phase and only those who voluntarily agreed were accepted. The patients filled in the questionnaire personally (paper-pen method) with a nurse specializing in diabetics present.

### 2.3. Research Tools

The questionnaire used in the research included basic socio-demographic data such as age, gender, education, place of residence, professional activity, marital status. Clinical data were collected from patients’ medical files. Standardized research instruments also included adherence in chronic diseases scale—ACDS—and Tilburg frailty indicator (TFI).

Adherence in Chronic Diseases Scale—ACDS—allows to evaluate the adherence with medical recommendations by adult patients suffering from chronic diseases. The presumption of ACDS lies in the fact that only high adherence reflects good therapeutic plan realization in respect to pharmacology. The scale consists of 7 questions with 5 possible answers each. The questions reflect behaviors directly determining adherence (questions 1–5) and the situations and opinions indirectly affecting adherence (questions 6–7). The result of ACDS ranges from 0–28 points. The higher the result, the higher the adherence. The score might be interpreted in the following way: above 26 points—high adherence, between 21–26 points—medium adherence, below 21 points—low adherence [41].

Non-adherent patient—a patient who achieved between 0–20 points in ACDS.

Adherent patient—a patient whose score was medium (21–26 points) or high (27–28 points in ACDS).

Tilburg Frailty Indicator (TFI)—is an instrument created by Gobbens et al. [42]. It allows the assessing of frailty syndrome reliably and globally. The unquestionable asset of this test is the fact that it not only evaluates physical determinants as other similar tools do, but it has some insight into psychological and social determinants as well. TFI consists of two parts. TFI includes a tool for identifying factors that determine weakness. It has been validated in five languages, including the Polish version [43]. The first part A consists of 10 questions about the participant’s socio-demographic characteristics. TFI includes a tool for identifying factors that determine weakness. Socio-demographic characteristics of age, sex, marital status and education, lifestyle, economic status, chronic diseases, stressful situations [44]. Part two B of the TFI comprises 15 self-reported questions, divided into three domains: physical, psychological, and social. The physical domain consists of eight questions related to physical health, unexplained weight loss, difficulty in walking, balance, hearing problems, vision problems, strength in hands, and physical tiredness (0–8 points). The psychological domain comprises four items related to cognition, depressive symptoms, anxiety, and coping (0–4 points). The social domain comprises three questions related to social relations, social support, and living alone (0–3 points). Eleven items of part two of the TFI have two response categories (“yes” and “no”), while the other items have three (“yes”, “no,” and “sometimes”). “Yes” or “sometimes” responses are scored 1 point each, while “no” responses are scored 0. The instrument’s total score may range from 0 to 15: the higher the score, the higher one’s frailty. Frailty is diagnosed when the total TFI score is ≥5 [40]. The TFI is valid and reproducible for the assessment of frailty syndrome among a Polish population. Cronbach’s alpha reliability coefficients of the instrument ranged from 0.68 to 0.72 [45].

BMI (body mass index)—was calculated as a person’s weight in kilograms divided by the square of height in meters. A high BMI can be an indicator of high body fatness. We classified BMI into following categories: normal body weight amounts for BMI 18.5–24.9 kg m^2^, overweight ranges from BMI 25.0–29.9 kg m^2^ and obesity is diagnosed if BMI > 30.0 kg m^2^. BMI is a commonly applied indicator of obesity as it highly correlates with a percentage of fatty tissue in children and adults. The prognosis of a fatty tissue percentage is dependent on age (higher in older people), gender (higher in men), and race [46].

### 2.4. Statistical Analysis

The analysis of quantitative variables was performed by calculating the mean, standard deviation, median, quartiles, minimum, and maximum. The analysis of qualitative variables was performed by calculating the number and percentage of the occurrences of each value. Comparison of the values of qualitative variables in the groups was performed using the chi-square test (with Yates’s correction for 2 × 2 tables) or the Fisher’s exact test where low expected frequencies appeared in the tables. On the other hand, the comparison of the values of quantitative variables in the two groups was performed using the Mann-Whitney test. In turn, the comparison of the values of quantitative variables in the three groups was performed using the Kruskal-Wallis test. Post-hoc analysis with Dunn’ test was performed to identify statistically significantly different groups after detecting statistically significant differences.

The multivariate analysis of the independent influence of many variables on the quantitative variable was performed using the logistic regression method. The results are presented in the form of values of the regression model parameters with a 95% confidence interval.

## 3. Results

### 3.1. Descriptive Data

The average age of the respondents amounted to 70.25 ± 6.70. The number of female to male patients was comparable—women (87; 49.71%) vs. men (88; 50.29%). Most of them were city residents (154; 88%), lived in a relationship (117; 66.86%) and completed secondary (80; 45.71%) or occupational education (44; 25.14%). The average duration of the diabetes treatment was 12.1 ± 8.52 years and the average number of medications taken was 8.07 ± 4.42. Most of the patients were on oral anti-diabetes medicines (106; 60.57%). The examinees were most frequently overweight (67; 38.29%) or of 1st-degree obesity (55; 31.43%). The most common co-existent diseases included hypertension (143; 81.71%) and kidney diseases (140; 80.00%). (Table 1).

### 3.2. The Prevalence of Frailty Syndrome and Level of Adherence to Medication in Type 2 Diabetes Elderly Patients

In the group of 175 patients, 101 (57.71%) displayed medium, 39 (22.29%) low, and 35 (20.00%)—high adherence. Frailty syndrome was diagnosed in as many as 140 out of 175 respondents (80.00%) (Table 2).

The average indicator of adherence to medication was 23.13 ± 3.72 and a TFI indicator amounted to 6.95 ± 2.75. The TFI in the physical domain was 3.68 ± 1.96, psychological domain 2.09 ± 0.93, and in the social domain 1.19 ± 0.75 on average (Table 3).

The respondents provided detailed answers to the questions contained in the TFI questionnaire concerning the following domains: physical, psychological, and social. A group of 63 respondents (36.00%) stated that they felt healthy in terms of their physical condition, 29 (16.57%) of the respondents lost more than 6 kg or more in the last 6 months or 3 kg in a month despite the lack of such intention. Our respondents experienced difficulties on a daily basis due to physical fatigue (121; 69.14%), difficulty walking (104; 59.43%), poor eyesight (100; 57.14%), lack of strength in the hands (70; 40.00%), poor hearing (55; 31.43%), difficulties in maintaining balance (53; 30.29%). The vast majority had problems with the psychological components of TFI—only 46 (26.29%) of the respondents had no problems with memory, 39 (22.29%) of the respondents did not experience a drop in mood over the last month, 18 (10.29%) of the respondents did not feel nervous over the last month and 43 (24.57%) were unable to cope well with the problems. In terms of the social components of TFI, the responses were as follows: a group of 145 (82.86%) of the respondents claimed that they received enough support from others, 143 (81.71%) indicated in the questionnaire that they missed the company of other people, and 35 (20.00%)—that they lived alone.

### 3.3. Socio-Demographic and Clinical Variables in Adherent and Non-Adherent Groups

Both adherent and non-adherent groups were similar in terms of age (*p* = 0.053), gender (*p* = 0.493), education (*p* = 0.457) marital status (*p* = 0.078) and place of residence (*p* = 0.787). They were also compliant as for analysed aspects of clinical condition (Table 4).

### 3.4. Frailty Syndrome vs. Adherence to Therapeutic Recommendations

The analysis of the correlation between adherence to medication and frailty syndrome revealed that the median of the overall score of TFI was significantly higher (*p* ˂ 0.001) in the low adherence group (Mdn = 9, Q1–Q3; 7–10 pts) than in medium (Mdn = 6, Q1–Q3; 5–9 pts) or high (Me = 6.00, Q1–Q3; 4.5–8 pts) adherence groups. The median of the scores in the physical domain of TFI was also significantly higher in the low adherence group (Mdn = 5, Q1–Q3; 4–6 pts) than in the medium (Mdn = 4, Q1–Q3; 2–5 pts) or high adherence groups (Mdn = 3, Q1–Q3; 2–4.5 pts). A similar relevant correlation was observed in the psychological domain of TFI (*p* = 0.034). However, no essential correlations were noticed between average scores in the social domain of TFI and the level of adherence (*p* = 0.339) (Table 5).

### 3.5. Non-Adherence Predictors vs. Multifactorial Regression Model

The multifactorial logistic regression model showed that there were three factors significant as independent predictors of the chance to be qualified to the non-adherence group, namely, TFI, male gender, and the number of all medications taken daily by a patient. Each point scored at the TFI scale increased the chance of being qualified to the non-adherence group by 55.8% OR 1.558, (95% CI 1.245–1.95). Being male increased the chance by 2.954 times in comparison to being female, OR 2.954, (95% CI 1.044–8.353). The odds ratio for all the medications taken daily by a DM2 elderly patient was OR 0.847–each extra tablet/pill decreased the non-adherence group qualification chance by 15.3% (95% CI 0.728–0.954) (Table 6).

## 4. Discussion

Adherence to medical and dietary recommendations, undertaking physical activity and self-checks are crucially important to avoid severe and chronic complications in diabetes, however, they may all appear hard to fulfill in the elderly in terms of deteriorating mental and physical abilities [47]. One of the primary examinations carried out in Poland in the group of elderly DM2 patients is focused on assessing relevant predictors of adherence to medication and co-existent frailty syndrome.

The results of this research revealed that most of the elderly DM2 patients suffered from frailty syndrome as well. It was also proved that the level of adherence to medication was significantly determined by the existence of the syndrome. Eventually, it was observed that the low adherence group of patients achieved significantly higher overall TFI scores than the medium and high adherence groups. They had higher indicators of TFI in the physical and psychological domains of TFI as well. The relevant independent predictors of non-adherence qualification chance were three factors such as TFI score, male gender, and the number of all medications taken by a patient daily.

The results confirm the frequent co-existence of type 2 diabetes and frailty syndrome in patients aged over 60 as in 80% of the respondents diagnosed in the study. Such a high percentage of the coexistence of frailty syndrome in these patients is disturbing. The meta-analysis of the 32 studies by Kong et al. (2021) shows that the pooled prevalence of frailty and prefrailty in older adults with diabetes was 20.1% (95% CI = 16.0–24.2%) and 49.1% (95%CI = 45.1–53.1%), respectively, with significant heterogeneity across the studies [48]. The result obtained by us differs from the one presented in the meta-analysis; however, it is worth noting that Kong et al. did not take into account any of the Polish studies. Another study (Survey of Health, Aging, and Retirement in Europe, SHARE) showed that more than 50% of the European population aged 50 years and over were pre-frail or frail [49]. However, here the age threshold of the respondents was much lower compared to our study, and besides, the authors did not focus only on diabetic patients but on the general population. The available literature basically lacks research similar to ours conducted in Poland. However, this does not change the fact that the common prevalence of frailty syndrome surely calls for special attention to be paid to the adherence with therapeutic recommendations in this group of patients.

Diabetes and frailty are two conditions that frequently occur concurrently and are increasingly prevalent in older patient [50]. Yoon et al. (2019) claim that frailty syndrome appears to be the third category of complications in elderly diabetes patients along with common microvascular diseases and complications which lead to serious disabilities [6]. Diabetic patients are more likely to be frail than non-diabetic older adults [10,51]. In the study by Chhetri et al. (2017) (*n* = 10, 039, mean age of 70.51) the prevalence of frailty syndrome among diabetic patients was higher compared to non-diabetic older adults (19.32% vs. 11.92%). In this study diabetics were at 1.36 (95% CI  =  1.18, 1.56) and 1.56 (95% CI  =  1.32, 1.85) fold increase in risk of frailty compared to non-diabetic population for prevalence and incidence respectively [10]. The research by Ferri-Guerra et al. (2019) showed that in 763 DM patients (mean age 72.9 years) 50.5% were frail [52]. Other studies also confirmed that the prevalence of frailty in adults older than 65 years was three- to five-fold higher in patients with diabetes than that in the general population [51,53].

Patients with diabetes mellitus are at risk for developing frailty due to the complex interplay between different cardiometabolic factors [54]. A 2013 study by Bouillon et al. lists adiposity, low high-density lipoprotein (HDL)-cholesterol level, high blood pressure, and cigarette smoking as the risk factors of frailty syndrome occurrence [53]. García-Esquinas et al. (2015) explained the association between diabetes mellitus and frailty by unhealthy behaviors and obesity and to a greater extent by poor glucose control and altered serum lipid profile among diabetic individuals [55].

The self-report study is consistent with other studies because it focused on assessing the degree of adherence to treatment and its connotation with the weakness syndrome in elderly patients with type 2 diabetes. The results showed that the majority of respondents showed moderate and poor compliance. A high degree of compliance with the recommendations was noted only in every fifth respondent. An analysis of the correlation between drug use and the severity of the fragile syndrome also showed that the mean TFI overall score was significantly higher in the low adhesion group compared to the moderate and high adherence patients. Likewise, the numbers in the physical and psychological domains were relatively higher in the low adherence group.

Prevalence of non-adherence among DM2 patients ranged between 6.9% and 90.6%. which may be due to differences in study designs methodology and populations [56]. Following therapeutic recommendations forces patients to change their existing lifestyle, requires constant education, developing the ability to properly interpret changes in their health condition, and coping with new situations. Therefore, if the physical domain of frailty syndrome (physical health, unintended body weight loss, walking and balance problems, hearing and vision impairment, fatigue, and grip power decrease) is to be analyzed in the context of adherence, it is easily recognizable that the patients experienced serious issues in the field of physical health. Reports from 2005–2007 by other researchers also prove rapid pathological changes in diabetics. The research by health, aging, and body composition (Health ABC) showed that elderly type 2 diabetes patients lose the power in the knee extensor muscles much earlier than non-diabetics [57]. The English reported that elderly men with newly diagnosed diabetes revealed much weaker muscle strength and a higher probability of physical functions impairment than non-diabetic patients [58]. In the study by Bourdel-Marchasson et al. (2019) the sarcopenia symptoms were found more often in patients with frailty syndrome and were connected with a decreased volume of grey matter responsible for locomotor control [59]. Diabetes also influenced the grip strength in the study by Gundmi et al. 2018 [60]. Based on the results of the self-report, patients can conclude that the lower the adhesion to drugs, the higher the value in the physical domain of TFI. Motta et al. (2020) indicated that muscle strength is probably the most important factor that can be improved by exercise in patients with brittleness syndrome [61].

Therefore, while planning education among such patients suffering additionally from diabetes, we suggest taking advantage of this conclusion and the findings. They may serve as a strong motivator encouraging this age group to physical activity. As diabetes mellitus negatively affects muscle system functioning, such interventions, aimed at improving muscle functioning, should be implemented as soon as possible and patients should be informed about them. Due to the description of the sample group and current COVID-19 situation adjusting a proper method of adherence control may appear problematic. Tele-medicine might be a good solution here. Telepharmacy increases adherence to therapeutic recommendations, especially when the system is enriched with the possibility of remote consultations between the patient and the pharmacist [62].

After the analysis of the self-reported results in the psychological domain of TFI, it might also be concluded that psychological components were at a higher level in the low adherence group in comparison to medium and high adherence groups. Earlier studies showed that DM2 patients are predestined to cognitive impairment and the changes affect adherence. For example, Tamura et al. (2018) showed that frailty and cognitive impairment are prevalent among patients with the cardiometabolic disease [63]. According to Munshi (2017), diabetes is a risk factor for the development of vascular as well as neurodegenerative dementia [64]. Considering what was mentioned above, patients with cognitive impairment may double medication doses or forget to take it, forget insulin injections and monitor blood glucose, as well as eat on time [19]. Apart from memory issues, half of the respondents in the study reported mood changes over the last month and the same number of them experienced nervousness. People with diabetes are two to three times more likely to develop depressive symptoms. Depressive symptoms are common in patients with uncontrolled type 2 diabetes who also have neuropathy and retinopathy. Hypertension, cardiovascular disease, and an unhealthy diet have been associated with depression [65,66].

In the other study of patients with DM2 (mean age 75.2 years old) carried out in Portugal, the group 22.3% had cognitive impairment, 16% had depression and 23.4% had anxiety. Those authors showed that higher anxiety and depression were associated with non-adherence to medication and to physical activity [19]. The analysis of the self-reported data in comparison to other studies confirms that planning proper management of diabetes mellitus in elderly patients should be based on the complex assessment of the patients in terms of frailty syndrome and its components (physical, psychological, and social). Frailty is the consequence of the interaction between the aging process and some chronic diseases and conditions that compromise functional systems [67].

Studies have shown that there are three factors of independent predictors of the chance of qualifying for non-adherence in the elderly, namely TFI score, male gender, and the number of medications taken daily by the patient. Self-report studies showed that each point obtained in the TFI scale increased the chance of qualifying a patient to the non-adherence group. Fragility negatively affected adherence by patients with other chronic diseases.

Another study from Poland shows that the coexistence of frailty syndrome has a negative impact on the adherence of older patients with hypertension [68]. In the study by Jankowska-Polańska et al. (2016), frail patients with hypertension had lower medication adherence in comparison to the non-frail subjects (6.60 ± 1.89 vs. 7.11 ± 1.42; *p* = 0.028) [69]. Frailty was associated with poor medication adherence in the study conducted among Chinese community-dwelling older patients with chronic diseases. In this study, authors concluded that medication necessity and medication concerns attenuated the total effect of frailty on medication adherence by 13.6% and 70.3% respectively [70]. The results above seem to confirm the opinion by Strain et al. (2021)—they are of the opinion that “frailty, rather than age, determines the prognosis for older adults with diabetes and should therefore be a key determinant of target setting and treatment choices when individualizing care” [71].

The self-reported study showed additionally that a significant predictor of non-adherence, except for TFI, was being male—it increased the chance of being qualified to the non-adherence group by 2954 times in comparison to being female. It was also found in the research of Horii et al. (2019) that being male was a significant predictor of adherence (OR 0.45, 95% CI 0.23–0.89, *p* = 0.022) [72]. It is worth mentioning that the impact of gender on adherence in DM2 patients was reported in existing studies heterogeneously. In the study by Demoz et al. (2020) predictors statistically associated with poor adherence were being female and the presence of at least one diabetic complication [73]. In the study by Aloudah et al. (2018) gender did not determine the level of adherence to oral hypoglycemic agents and lower adherence was associated with younger age, higher numbers of non-oral hypoglycemic agents, and higher HbA1c levels [74]. Gender did not influence adherence in the research by Aminde et al. (2019) either. In multivariable analysis, authors showed that an age above 60 years, alcohol consumption, and insulin-only therapy were associated with non-adherence [20].

The third aspect which affected adherence in the research was the number of medications taken by a patient daily. The odd ratio for all the medications taken daily was OR 0.847 and each extra tablet decreased the chance of being qualified to the non-adherence group by 15.3%. The method of DM treatment and the number of diabetic medications did not determine adherence.

Polypharmacy has previously been shown to be inversely associated with medication adherence in several studies of patients with DM2 [74,75] and in studies of patients with other chronic diseases [76,77]. The self-reported findings in correlation with the results by other researchers suggest reducing the number of medications to the most necessary ones to maintain optimal health conditions in over 60 years old with DM2, co-existent diseases, and frailty syndrome. It seems to be one of the non-adherence prophylaxis methods. The self-reported study also indicated the factors of non-adherence in over 60 years old DM2 patients. The similarities and differences between the results of the study and the results achieved by other researchers in terms of non-adherence predictors indicated a need for more personalized drug selection and therapeutic management to improve clinical outcomes in this group. It is important to notice that a lot of research cited in the discussion above is concentrated on adherence in respect to DM2, elderly age or frail patients with other chronic diseases. It stems from the deficits in literature on the topic of adherence, DM2, old age and frailty syndrome.

We suggest preceding the stage of planning proper diabetes management with a complex assessment of DM elderly patients for the occurrence of frailty syndrome and its components (physical, psychological, and social) with the use of a standardized instrument. We also suggest repeating the assessment in regular intervals or when needed to optimize health care. Moreover, there is a need for more personalized drug selection and therapeutic management to improve clinical outcomes in this group. The complex assessment of DM elderly patients for the occurrence of frailty syndrome and its components will allow members of the therapeutic team (doctors, nurses, social workers, nutritionists, rehabilitators) to jointly develop and implement individualized interventions aimed at the patient’s needs to improve his adherence to medication. Rather, the focus should be on preventing non-adherence at an early stage of the disease by taking steps in the right order, i.e., (1) assessing the presence of the patient’s weakness syndrome, (2) undertaking interventions aimed at the patient’s physical, psychological, or social sphere depending on the assessment result, aimed at optimizing his adherence, (3) evaluating the medication adherence and glycemic control. However, checking the effectiveness of such a solution requires further research.

The first crucial limitation of the study derives from a small sample group analyzed and the limitation to one voivodship. The second is that only a questionnaire was used to assess adherence. The use of an analysis of pharmacy registers and medication use control in addition to the questionnaire would significantly increase the clinical value of the analysis however as Denicolò et al. (2021) notice there is no golden standard to assess adherence [56]. The third limitation referred to the ACDS questionnaire which only examined adherence in terms of pharmacotherapy, while in DM patients in the context of adherence, there are other elements equally essential such as regular physical activity, adherence with dietary recommendations, regular blood pressure, and glycemia measurements, etc.

## 5. Conclusions

Frailty syndrome was found in four-fifths of the DM2 elderly patients and it affected the adherence to medication in this group. The low adherence group achieved higher overall TFI scores and separately higher ones in physical and psychological domains in comparison to medium and high adherence groups. The independent predictors affecting non-adherence included three factors: a TFI score, male gender, and the number of all medications taken by a patient daily.

## Figures and Tables

**Figure 1 jcm-11-01707-f001:**
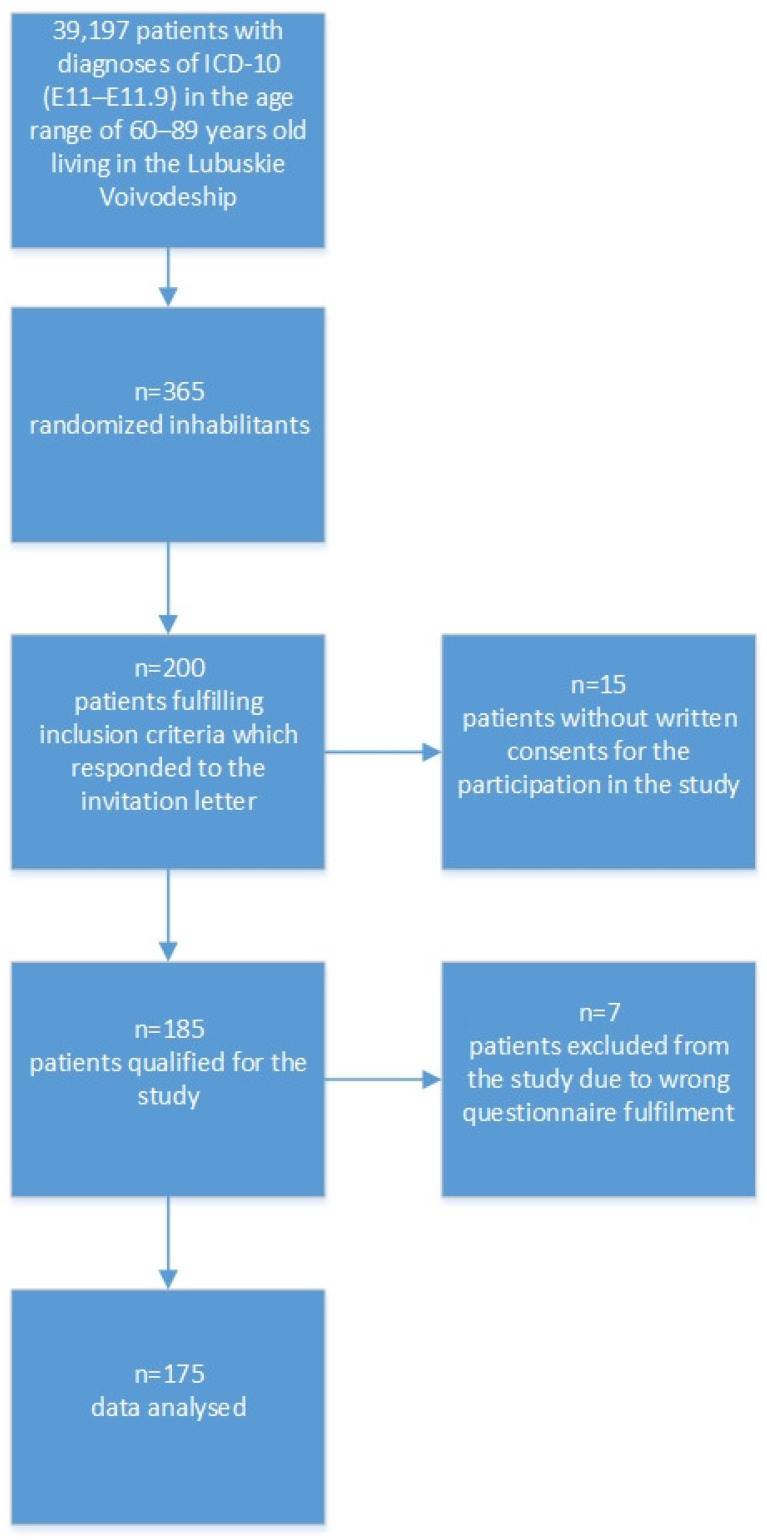
Sampling after considering research inclusion and exclusion criteria.

**Table 1 jcm-11-01707-t001:** Socio-demographic and clinical characteristics of the sample group.

Variables		Total (*n* = 175)
Gender	Women	87 (49.71%)
	Men	88 (50.29%)
Age	M ± SD	70.25 ± 6.70
	median	69
	Q1–Q3	65–74
Marital status	Single	58 (33.14%)
	In relationship	117 (66.86%)
Place of residence	Village	21 (12.00%)
	City	154 (88.00%)
Education	Primary	19 (10.86%)
	Occupational	44 (25.14%)
	Secondary	80 (45.71%)
	High	32 (18.29%)
Diabetes duration [years]	M ± SD	12.1 ± 8.52
	median	10
	Q1–Q3	5–15
Number of all daily medications	M ± SD	8.07 ± 4.42
	median	7
	Q1–Q3	6–10
Number of anti-diabetes medications taken daily	M ± SD	1.81 ± 1.25
	median	2
	Q1–Q3	1–3
Diabetes treatment method	Oral anti-diabetes medications	106 (60.57%)
	Insulin	31 (17.71%)
	Oral anti-diabetes medications and insulin	33 (18.86%)
	non-pharmacological methods	5 (2.86%)
Body Mass Index (BMI)	normal body weight	19 (10.86%)
	overweight	67 (38.29%)
	1st degree obesity	55 (31.43%)
	2nd/3rd degree obesity	34 (19.43%)
Co-existent diseases: Hypertension	No	32 (18.29%)
	Yes	143 (81.71%)
Co-existent diseases: ischaemic heart disease	No	113 (64.57%)
	Yes	62 (35.43%)
Co-existent diseases: Rheumatic diseases	No	130 (74.29%)
	Yes	45 (25.71%)
Co-existent diseases: Kidney’s diseases	No	140 (80.00%)
	Yes	35 (20.00%)
Co-existent diseases: Respiratory system diseases	No	137 (78.29%)
	Yes	38 (21.71%)
Co-existent diseases: Locomotor system disorders	No	121 (69.14%)
	Yes	54 (30.86%)
Co-existent diseases: Diabetic foot syndrome	No	137 (78.29%)
	Yes	38 (21.71%)
Co-existent diseases: Eye diseases	No	103 (58.86%)
	Yes	72 (41.14%)

M—mean; SD—standard deviation, Q1—first quartile; Q3—third quartile; ACDS, adherence in chronic diseases scale.

**Table 2 jcm-11-01707-t002:** Frailty syndrome occurrence and level of adherence to medication.

Instrument	Points	Interpretation	*n*	%
TFI	0–4	lack of frailty syndrome	35	20.00%
	5 and more	frailty syndrome	140	80.00%
ACDS	0–20	low adherence	39	22.29%
	21–26	medium adherence	101	57.71%
	27–28	high adherence	35	20.00%

TFI—Tilburg frailty indicator; ACDS—adherence in chronic diseases scale.

**Table 3 jcm-11-01707-t003:** Average distribution measures for standardized instruments.

Instrument	N	M	SD	Mdn	Min	Max	Q1	Q3
ACDS	175	23.13	3.72	24	13	28	21	26
TFI total score	175	6.95	2.75	7	0	13	5	9
TFI: physical domain	175	3.68	1.96	4	0	8	2	5
TFI: psychological domain	175	2.09	0.93	2	0	4	2	3
TFI: social domain	175	1.19	0.75	1	0	3	1	2

TFI, Tilburg frailty indicator; ACDS, adherence in chronic diseases scale; M, mean; SD, standard deviation; Mdn, median; Q1, first quartile; Q3, third quartile.

**Table 4 jcm-11-01707-t004:** Socio-demographic and clinical variables in adherent and non-adherent groups.

Variables		ACDS		*p*
		Adherent (*n* = 136)	Non-Adherent (*n* = 39)	
Age	M ± SD	69.64 ± 6.27	72.38 ± 7.72	U = 2113.5 *p* = 0.053
	Me	69	70	
	Q1–Q3	65–73	67.5–76	
Gender	Women	70 (51.47%)	17 (43.59%)	chi2 = 0.471 *p* = 0.493
	Men	66 (48.53%)	22 (56.41%)	
Marital status	Single	40 (29.41%)	18 (46.15%)	chi2 = 3.116 *p* = 0.078
	In relationship	96 (70.59%)	21 (53.85%)	
Education	Primary	14 (10.29%)	5 (12.82%)	*p* = 0.457
	Occupational	31 (22.79%)	13 (33.33%)	
	Secondary	64 (47.06%)	16 (41.03%)	
	High	27 (19.85%)	5 (12.82%)	
Place of residence	Village	16 (11.76%)	5 (12.82%)	*p* = 0.787
	City	120 (88.24%)	34 (87.18%)	
Diabetes duration [years]	M ± SD	11.88 ± 8.14	12.87 ± 9.81	U = 2576 *p* = 0.786
	Me	10	10	
	Q1–Q3	15–5	5–18.5	
Number of anti-diabetes medications taken daily	M ± SD	1.84 ± 1.3	1.69 ± 1.08	U = 2788.5 *p* = 0.609
	Me	2	2	
	Q1–Q3	1–3	1–2	
Number of all daily medications	M ± SD	8.11 ± 4.7	7.92 ± 3.32	U = 2524 *p* = 0.645
	Me	7	8	
	Q1–Q3	5–10.25	6–10	
Body Mass Index (BMI)	normal body weight	15 (11.03%)	4 (10.26%)	*p* = 0.601
	overweight	49 (36.03%)	18 (46.15%)	
	1st degree obesity	43 (31.62%)	12 (30.77%)	
	2nd/3rd degree obesity	29 (21.32%)	5 (12.82%)	
Co-existent diseases: Hypertension	No	26 (19.12%)	6 (15.38%)	chi2 = 0.088 *p* = 0.767
	Yes	110 (80.88%)	33 (84.62%)	
Co-existent diseases: ischaemic heart disease	No	90 (66.18%)	23 (58.97%)	chi2 = 0.408 *p* = 0.523
	Yes	46 (33.82%)	16 (41.03%)	
Co-existent diseases: Rheumatic diseases	No	103 (75.74%)	27 (69.23%)	chi2 = 0.374 *p* = 0.541
	Yes	33 (24.26%)	12 (30.77%)	
Co-existent diseases: Kidney’s diseases	No	111 (81.62%)	29 (74.36%)	chi2 = 0.596 *p* = 0.44
	Yes	25 (18.38%)	10 (25.64%)	
Co-existent diseases: Respiratory system diseases	No	109 (80.15%)	28 (71.79%)	chi2 = 0.801 *p* = 0.371
	Yes	27 (19.85%)	11 (28.21%)	
Co-existent diseases: Locomotor system disorders	No	95 (69.85%)	26 (66.67%)	chi2 = 0.034 *p* = 0.855
	Yes	41 (30.15%)	13 (33.33%)	
Co-existent diseases: Diabetic foot syndrome	No	109 (80.15%)	28 (71.79%)	chi2 = 0.801 *p* = 0.371
	Yes	27 (19.85%)	11 (28.21%)	
Co-existent diseases: Eye diseases	No	81 (59.56%)	22 (56.41%)	chi2 = 0.028 *p* = 0.867
	Yes	55 (40.44%)	17 (43.59%)	
Diabetes treatment method	Oral anti-diabetes medications	81 (59.56%)	25 (64.10%)	*p* = 0.683
	Insulin	23 (16.91%)	8 (20.51%)	
	Oral anti-diabetes medications and insulin	27 (19.85%)	6 (15.38%)	
	non-pharmacological methods	5 (3.68%)	0 (0.00%)	

*p*—for quantitative variables the Mann-Whitney test, for qualitative variables chi-square test or Fisher’s exact test, M—mean, SD—standard deviation, Me—median, Q1—first quartile, Q3—third quartile, ACDS—adherence in chronic diseases scale.

**Table 5 jcm-11-01707-t005:** The correlation between adherence to medication and frailty syndrome.

Tilburg Frailty Indicator (TFI)		ACDS			*p*
		Low Adherence—A (*n* = 39)	Medium Adherence—B (*n* = 101)	High Adherence—C (*n* = 35)	
TFI total score	M ± SD	8.62 ± 2.27	6.64 ± 2.69	6 ± 2.72	*p* < 0.001 * A > B, C
	Mdn	9	6	6	
	Q1–Q3	7–10	5–9	4.5–8	
TFI: physical domain	M ± SD	4.87 ± 1.64	3.43 ± 1.94	3.09 ± 1.82	*p* < 0.001 * A > B, C
	Mdn	5	4	3	
	Q1–Q3	4–6	2–5	2–4.5	
TFI: psychological domain	M ± SD	2.44 ± 0.99	2.02 ± 0.91	1.89 ± 0.83	*p* = 0.034 * A > B, C
	Mdn	2	2	2	
	Q1–Q3	2–3	2–2	1–2	
TFI: social domain	M ± SD	1.31 ± 0.92	1.2 ± 0.69	1.03 ± 0.66	*p* = 0.339
	Mdn	1	1	1	
	Q1–Q3	1–2	1–2	1–1	

*p*—Kruskal-Wallis test and post-hoc analysis (Dunn’s test), * statistically significant relationship (*p* < 0.05), TFI, Tilburg frailty indicator; M, mean; SD, standard deviation; Me, median; Q1, first quartile, Q3, third quartile.

**Table 6 jcm-11-01707-t006:** Non-adherence predictors-multivariate logistic regression model.

Variables		OR	95% CI		*p*
Tilburg Frailty Indicator (total score)		1.558	1.245	1.95	<0.001 *
Age	(years)	1.062	0.98	1.152	0.144
Gender	Women	1	ref.		
	Men	2.954	1.044	8.353	0.041 *
Marital status	Single	1	ref.		
	In relationship	0.524	0.175	1.565	0.247
Education	Primary	1	ref.		
	Occupational	3.609	0.562	23.184	0.176
	Secondary	0.854	0.155	4.717	0.856
	High	1.11	0.132	9.328	0.923
Place of residence	Village	1	ref.		
	City	0.554	0.125	2.448	0.436
Body Mass Index (BMI)	normal body weight	1	ref.		
	overweight	1.839	0.374	9.048	0.454
	1st degree obesity	1.747	0.347	8.786	0.499
	2nd/3rd degree obesity	0.728	0.121	4.375	0.728
Co-existent diseases: Hypertension	No	1	ref.		
	Yes	3.111	0.76	12.729	0.114
Co-existent diseases: Ischemic heart disease	No	1	ref.		
	Yes	1.271	0.476	3.399	0.632
Co-existent diseases: Rheumatic diseases	No	1	ref.		
	Yes	2.223	0.667	7.407	0.193
Co-existent diseases: Kidney’s diseases	No	1	ref.		
	Yes	1.946	0.553	6.854	0.3
Co-existent diseases: Respiratory system diseases	No	1	ref.		
	Yes	1.119	0.361	3.471	0.846
Co-existent diseases: Locomotor system disorders	No	1	ref.		
	Yes	0.926	0.324	2.646	0.886
Co-existent diseases: Diabetic foot syndrome	No	1	ref.		
	Yes	0.384	0.106	1.391	0.145
Co-existent diseases: Eye diseases	No	1	ref.		
	Yes	1.149	0.402	3.285	0.796
Diabetes duration [years]	[years]	0.97	0.908	1.035	0.358
Diabetes treatment method	Oral anti-diabetes medications	1	ref.		
	Insulin	0.778	0.119	5.086	0.793
	Oral anti-diabetes medications and insulin	1.112	0.281	4.392	0.88
	non-pharmacological methods	0	0	Inf	0.991
Number of anti-diabetes medications taken daily		0.78	0.401	1.514	0.462
Number of all daily medications		0.847	0.728	0.984	0.03 *

*p*—multivariate logistic regression, * statistically significant relationship (*p* < 0.05), OR, odds ratio, ref.—reference category.

## Data Availability

Data confirming the reported results can be found at the Department of Nursing of the University of Zielona Góra. Responsible person: Iwona Bonikowska.

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
