# Peer review of "Adherence to Medication in Older Adults with Type 2 Diabetes Living in Lubuskie Voivodeship in Poland: Association with Frailty Syndrome"

_jcm, 2022, doi:10.3390/jcm11061707_

Round 1
Reviewer 1 Report
I believe that the topic of the manuscript is interesting. However, this manuscript does not seem to present any meaningful research results. Please refer to the following review.
Specific comments:
Most of variables in TFI were found to be insignificant at Table 5 in this study. Overall please revise the research design and analysis.
Please indicate whether the authors’ consent was obtained for the research tools used in this study.
Please present the reference number of Gobbens et al. on page 5. In general, please check whether the reference number is omitted in the text and enter it.
Please describe the TFI in more detail, such as explanation of scale levels et al.
In the statistical analysis, the terms which are quantitative variables and qualitative variables are not accurate. Please revise the terms.
Please present values of chi-square test on Table 1. In Table 1, meanings of median, Q1, Q3 are confusing to understand. Generally the contents in Table 1 are very confusing. Please correct variables and values consistently.
Please present values of inference statistics in Tables 2 and 3.
There are some old references in your manuscript. Replace them with references within the last five years and also add DOI to your references. Please rewrite the references by referring to the writing guidelines.
Author Response
The autors thank you for this comment.

Reviewer 2 Report
The study was carried out among DM2 patients. The independent predictors of the chance to be qualified to the non-adherence group included three indicators: Tilburg Frailty Indicator, male gender and the number of all medications taken daily. Weakness syndrome in elderly DM2 patients influenced medical adherence in this group.
Line 183: normal body weight … BMI 18.5-25.9 must be corrected with BMI 18.5 - 24.9
Line 231: …19 + 0.75 must be corrected with 1.9 …
Line 317: reference 13 must be corrected with 11
Line 436: Hotii must be corrected with Horii
References section must be standardized: punctuation between name and surname (correct n.2); number of Authors to be included (correct n.8, 44, 45 ..); initial and final page (correct n.1, 10, 14 ...)
Author Response
Autorzy dziękują za ten komentarz.

Reviewer 3 Report
Introduction: The authors include excess unnecessary information, on the other hand, the justification is fragile, as the innovative contribution of the study is not clear. It is necessary to describe a summary of the results of similar studies and highlight the differential of this research.
Methods:
- Improve the resolution of figure 1.
Results:
- Remove subtitles
Discussion:
- Remove subtitles
- The first two paragraphs repeat results. I suggest that the authors only briefly cite the results, showing the discussion of each finding right after.
- Throughout the discussion, the authors repeat many times and in great detail the results mentioned above.
Conclusion:
- The practical implications should be detailed at the end of the discussion and not as a separate subtitle at the conclusion of the study.
Author Response
The autors thank you for this comment.

Round 2
Reviewer 1 Report
Although the authors have corrected this manuscript, I am still confused. The sample sizes of non-adherent patients and adherent patients are not presented. Why is the non-parametric test applied in Table 4. What is the table on page 12. Therefore, it is difficult for me to acknowledge the reliability of the results of this manuscript.Author Response
The authors of the publication would like to thank your valuable comments, which will be in the future to create good-quality research. We hope that the results of our research will help to improve the effectiveness of therapy in elderly patients with type 2 diabetes. Thank yor very much.

Reviewer 3 Report
Consider that the paper is ready for publication
Author Response
The autors of the publication would like to thank you for your valuable comments, which will be used in the future to create good-quality research. We hope that the results of our research will help to improve the effectiveness of terapy in elderly patients with typ 2 diabetes. Thank yor very much.